# Superheated Steam Torrefaction of Biomass Residues with Valorisation of Platform Chemicals—Part 1: Ecological Assessment

**Baharam Roy [1], Peter Kleine-Möllhoff [2,\*] and Antoine Dalibard [3]**

[1] Reutlingen Research Institute (RRI), Reutlingen University, 72762 Reutlingen, Germany;
Md_Baharam.Sk@Reutlingen-University.DE or roybaharam@gmail.com

[2] ESB Business School, Reutlingen University, 72762 Reutlingen, Germany

[3] Fraunhofer Institute for Interfacial Engineering and Biotechnology IGB, 70569 Stuttgart, Germany;
antoine.dalibard@igb.fraunhofer.de

**\*** Correspondence: peter.kleine-moellhoff@reutlingen-university.de; Tel.: +49-712-1271-5009

**Abstract:** Within the last decade, research on torrefaction has gained increasing attention due to its ability to improve the physical properties and chemical composition of biomass residues for further energetic utilisation. While most of the research works focused on improving the energy density of the solid fraction to offer an ecological alternative to coal for energy applications, little attention was paid to the valorisation of the condensable gases as platform chemicals and its ecological relevance when compared to conventional production processes. Therefore, the present study focuses on the ecological evaluation of an innovative biorefinery concept that includes superheated steam drying and the torrefaction of biomass residues at ambient pressure, the recovery of volatiles and the valorisation/separation of several valuable platform chemicals. For a reference case and an alternative system design scenario, the ecological footprint was assessed, considering the use of different biomass residues. The results show that the newly developed process can compete with established bio-based and conventional production processes for furfural, 5-HMF and acetic acid in terms of the assessed environmental performance indicators. The requirements for further research on the synthesis of other promising platform chemicals and the necessary economic evaluation of the process were elaborated.

**Keywords:** biorefinery; superheated steam torrefaction; environmental assessment; volatile recovery; platform chemicals



## 1. Introduction

What would the world look like if we could realise the production of chemical products in a bio-based circular economy? What impact would bio-based chemistry have on the climate and the environment? How can the torrefaction of biomass be used for environmentally friendly chemicals production?

### 1.1. General Context

Sustainable biomass will play a significant role in meeting the 2030 target to reduce greenhouse gas emissions, as well as the objective of climate neutrality by 2050 in the European Green Deal [1,2]. Biomass can be used as a renewable energy source, a material substitute and as a carbon sink, thus contributing towards negative emissions. A recent study shows that the current trend in EU biomass use has to be corrected in order to achieve a net zero economy. Traditional bioenergy applications will become less and less competitive due to new future options based on increasing electrification and hydrogen share. Instead, material uses of biomass must increase significantly with a special focus on high-value applications (chemicals, textiles, etc.). The use of biomass for energy application must be reserved for special niches (e.g., industrial heat, fuels for aviation, etc.) [3]. Therefore, innovative technologies that allow for this strategic change are strongly required.

Bio-based products can be used in small, specialised but also large-volume markets. In the area of fine and speciality chemicals as well as active pharmaceutical ingredients, bio-based products are already competitive to some extent due to their functionality and thus offer worthwhile investment targets [4]. Biddy et al. (2016) presented a study in which 12 promising chemicals were identified that can already be produced in the near future from renewable sources, such as sugar, lignocellulose or algae [5]. For example, 5-hydroxymethylfurfural (HMF) and furfural are two interesting platform chemicals for a bio-based chemical production economy, and their current bio-based production has been studied in terms of their environmental footprint [5,6]. Some industrial companies in the chemical industry have already adapted their business model to a bio-based circular economy [7–10].

*1.2. Biomass Torrefaction*

In this context, biomass torrefaction is a promising technology since it allows one to upgrade low-quality biomass to higher quality products to be used either directly or further processed into high-value products [11,12]. Up to now, torrefied biomass is essentially used for energy applications, either directly for electricity generation (co-firing with coal in power plant) or as feedstock for further conversion into high quality biofuels (pyrolysis, gasification, catalytic synthesis, etc.). The increasing number of scientific investigations within the last decade in this field focused mainly on improving the energy density of biomass, the production of synthetic fuels and the integration into existing production and industrial structures for cascaded use [13–15]. In particular, the research works focused on application areas such as agriculture and food, the paper industry, energy suppliers, the steel industry, but also on the production of pyrolysis products, as well as liquefaction and gasification [4,11,15–18].

Torrefaction is a mild form of pyrolysis in which biomass is usually heated to about 200–300 °C. During the process, the three main constituents of woody biomass (cellulose, hemicellulose and lignin) are thermally decomposed at different degrees, leading to the formation of non-condensable gases (CO, $CO_2$) and condensable volatiles [11,18,19]. Beside water, the condensable volatile fraction contains valuable chemical substances (e.g., furfural, acetic acid, methanol, formic acid) that can be used as platform chemicals [19]. In this regard, the use of superheated steam as torrefaction agent is very interesting because it allows for a fast and uniform process and an easy recovery of volatiles [20–22]. In addition, the condensation heat can be recovered in order to make the whole process more energy efficient.

However, the production of value-added chemicals from torrefaction condensate is a challenging task due to the low concentrations and the resulted complex mixture of water, aldehydes, carboxylic acids, furans, ketones and alcohols. The organic fraction comprises innumerable substances with extensive distribution of molecular weight and polarity, which affects the effective separation of chemicals [23]. Therefore, new concepts are required in order to achieve an economical valorisation of these chemicals at the market-required purity.

For the reasons mentioned above, the authors of the present paper have dealt with the recovery and separation of valuable platform chemicals from torrefied biomass. In the context of this work, the ecological footprint of the newly developed process is investigated, which can be used to obtain valuable platform chemicals from biomass residues by torrefaction using superheated steam under atmospheric conditions. This analysis is intended to provide a statement on the ecological footprint of the newly developed process and how it compares to competing processes.

## 2. Superheated Steam Torrefaction
*2.1. History*

The core technology of the developed biorefinery concept is the superheated steam (SHS) processing under atmospheric pressure. This technology has been originally devel-

oped for drying purposes in the early 1990s [24], then further developed by Fraunhofer IGB for torrefaction within the EU project SteamBio (https://www.steambio.eu/ (accessed on 6 December 2021)). The use of SHS and thus the absence of oxygen permits an inert processing, prevents oxidation of the product and significantly reduces the risk of explosion.

## 2.2. Process Principle

The process principle is based on a system, which is hermetically closed at the top and atmospherically open at the bottom. The material to be processed is introduced to an SHS atmosphere, which is maintained in a superheated state through the supply of heat (see Figure 1 below). Enhanced heat transfer is achieved convectively by recirculating the SHS in a closed loop. The vapours (moisture, volatiles) released from the material during the thermal processing are extracted/condensed in order to maintain the system atmospheric pressure. The energy contained in the excess vapours (at temperature above 100 °C) can be recovered in other processes of the plant, which results in a high overall energy efficiency. Energy recovery can be conducted, e.g., by means of condensation, which allows volatile organic compounds (VOCs) to be condensed out with the excess steam. These condensable organic substances can be further separated and valorised as value-added products.

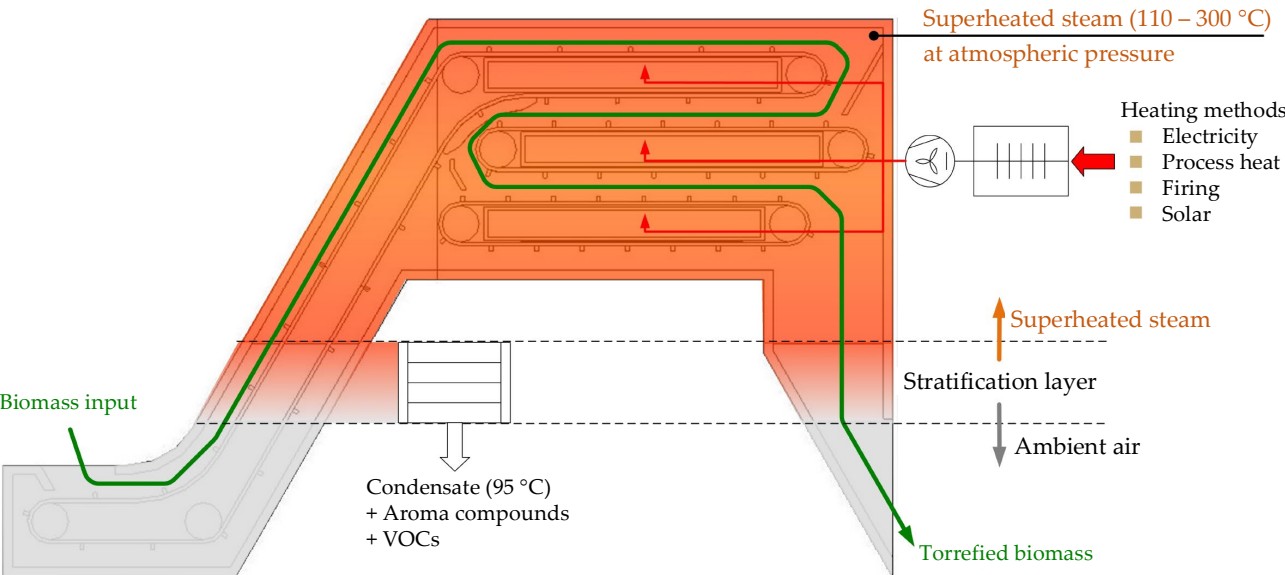

**Figure 1.** Principle design concept of the SHS drying and torrefaction process.

## 3. Materials and Methods

Within this paper, a new process for extracting valuable products from biomass that have been steam dried and torrefied under ambient pressure is analysed. This study is based on experimental results obtained within a German public-funded research project on an innovative biorefinery concept (see funding section). The experiments were conducted in a pilot-scale SHS-drying/torrefaction unit and laboratory-scale batch rectification and extraction set-ups.

### 3.1. Developed Refinery Concept

Figure 2 shows the different steps of the developed biorefinery concept, which was investigated in terms of its environmental impacts, as presented in this paper.

The process units with a grey background represent the reference case. In addition to this reference case, another variant was investigated in which the torrefied biomass is fed to an incineration plant (hatched background and dashed lines) that serves to supply heat internally to the biorefinery process and externally to potential consumers.

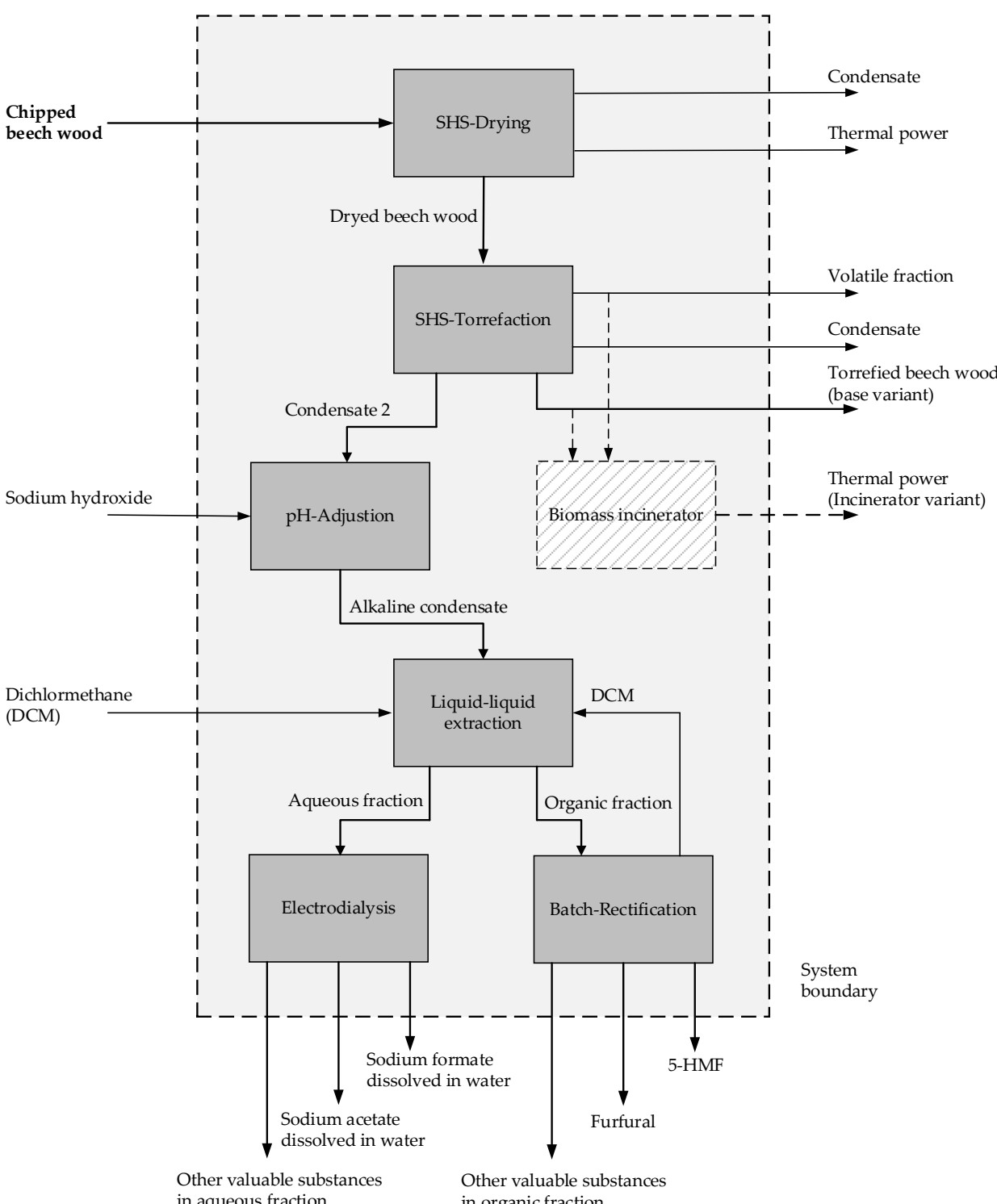

**Figure 2.** Biorefinery process set up.

### 3.2. Methods, Framework and Assumptions

Figure 3 shows the basic methodology used in this work. For each process step, the energy and material balances were drawn up on the basis of thermodynamic and fluid mechanics principles as well as experimental results. These balances formed the basis for the assessment of the environmental impact. The environmental impacts were carried out in accordance with ISO 14040 and 14044.

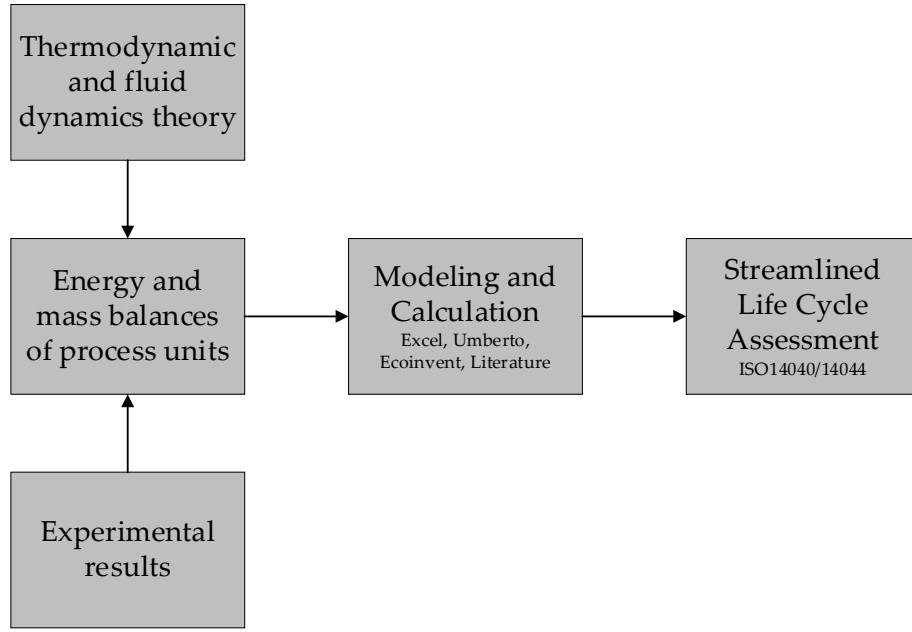

**Figure 3.** Assessment methodology.

The system boundary is shown in Figure 2. The upstream supply chains for biomass, energy and auxiliary materials were considered for the analysis. Table 1 lists the assumptions and parameters for the ecological analysis that allow for comparison with other relevant studies, such as [6,11,25].

**Table 1.** Assumptions and parameter for the ecological analysis.

| Designation | Value | Unit | Description |
|---|---|---|---|
| Functional unit | 1 | kg | Biomass input |
| Design input | 1000 | kg/h | Biomass input |
| Allocation | | | Mass related |
| Plant location | | | Germany |
| Base year | 2019 | | Before COVID-19 |
| External energy supply | | | Electricity and natural gas |
| Reference biomass type | | | Shredded beechwood |
| Sensitivity analysis | 20–45 | % (dry basis) | Moisture content biomass input |
| | 150–600 | kgCO$_2$/kWh | Specific CO$_2$-emission electricity |
| | 22–28 | MJ/kg | Higher heating value torrefied biomass |
| | 0 and 5 | % | Steam loss (referred to evaporated water) |
| | 100–300 | km | Biomass transportation distance |
| | Straw | / | Biomass input alternative |

In total, 1000 kg/h of biomass input material was chosen for the size of the plant and one kilogram of processed biomass as the functional unit to maintain the comparability with other studies. The allocation of the primary energy demand and CO$_2$ emissions to the different products was nevertheless also investigated.

The process structure with the energy and material flows was first calculated, then used as input for the modelling in the life cycle assessment software Umberto and supplemented with life cycle assessment inventory data obtained, e.g., from the Ecoinvent 3.7.1 database. The environmental impact assessment was carried out using the ReCiPe midpoint (H) w/o LT method [26]. For the reference case and the process variant, the impact categories climate change and primary energy consumption were considered. In addition, for two

different biomass inputs, the impact categories human toxicity, freshwater ecotoxicity and freshwater eutrophication were identified as significant and therefore considered. Germany was assumed as the location for the production plant. For the manual calculations and verification of the results from the Umberto software, primary energy factors and specific $CO_2$ emissions values for Germany from 2019 were used because the years 2020 and 2021 are not representative due to the COVID-19 pandemic. Nevertheless, the influence of different specific $CO_2$ emissions of electricity generation was investigated and presented to allow for a transfer of the results to other locations or other energy supply structures.

The incineration plant of the analysed variant was considered as a system extension in order to be able to present environmental credits in terms of primary energy demand and $CO_2$ emissions. A sensitivity analysis was carried out with regard to the factors listed in Table 1. The results obtained were compared and evaluated with available information from competing processes.

The considerations end at the gate of the process under investigation. This means that product transports or the onward transmission of the generated thermal energy were not considered. The research in this paper does not include any economic considerations, which will be evaluated in a future paper.

## 4. Results

### 4.1. Energy and Material Flows

Figure 4 shows the energy and material flows of the reference case for an assumed steam loss of 5% (related to the evaporated mass) during SHS drying and torrefaction.

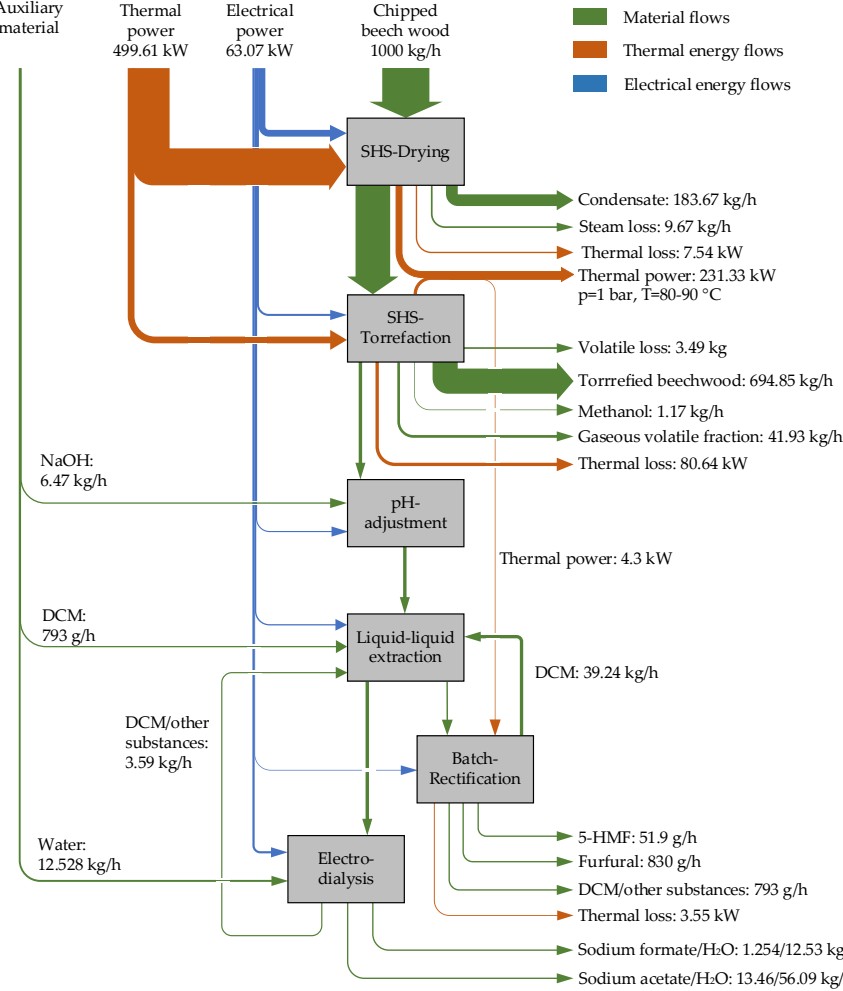

**Figure 4.** Energy and material input and output flows (reference case, 5% steam loss).

The width of the arrows in the Sankey diagram is not scaled exactly. They are only intended to illustrate which flows are significant. Therefore, the absolute amounts of the flows are indicated. The thermal and electrical energy flows are shown separately. The heat losses through the walls of the individual apparatus are not shown. Heat losses of 5% were assumed for the calculations. The energy flows due to the chemical internal energy of the materials fed into and removed from the system, i.e., in particular the woodchips and the torrefied biomass, are not shown in the Sankey diagram. In the Sankey diagram, the energy flow rates are given in kW, that is, kJ/s. All data refer to a thermal energy supply with natural gas, a chipped beech wood input flow of 1000 kg/h and 35 weight % (dry) water content of the biomass input for a typical production environment. In the following, the term "energy demand" is used for ease of reading when referring to the percentage shares of the individual energy flow rates.

The thermal energy demand of the dryer, which is operated at a temperature range between 150 and 180 °C, represents the largest energy demand, with 92.65% of the total energy demand, followed by the torrefaction unit, with 6.27%. The thermal energy demand of the rectification column is very small in comparison and can be covered internally from the torrefaction. The discharge of the non-condensable gaseous volatile fraction causes the most thermal losses in the torrefaction unit. The thermal output stream from the SHS-drying plant due to the condensation of the steam expelled from the biomass accumulates at a temperature level of approx. 95 °C (see Figure 1) and can be utilised for external purposes.

The fan that circulates the steam in the SHS-drying unit has the greatest demand for electrical energy, with 9.23% of the total energy consumption, followed by the fan of the torrefaction unit, with 0.69%. The electrodialysis unit consumes 2.78% of the total electrical energy. All other electrical energy consumers together require 10.90% of the total energy consumption.

In terms of mass flow quantity, with an input of 1000 kg/h of beech wood chips, the torrefied biomass represents the largest recyclable material output, with almost 70%, followed by the condensate from the dryer (approximately 18%), which can be thermally utilised.

The torrefied biomass has a higher heating value (HHV), approx. 22 MJ/kg, and can be utilised either as a very clean fuel with coal-like characteristics, as a raw material for further use in activated carbon production or for the production of synthesis gas. The non-condensable volatile fraction ($CO$, $CO_2$, $CH_4$, $H_2$) from torrefaction cannot be further utilised and is released to the environment in the reference case. The valuable substances 5-HMF, furfural, sodium formate and sodium acetate are produced in small quantities but high purities (>95%). The pilot tests have shown that about 20 different chemical substances can be extracted. However, only the above-mentioned platform chemicals are shown here, as they represent the largest share in terms of quantity. Dichloromethane is used as extraction agent and can be recovered up to 98% in the rectification unit.

The data presented up to this point are based on a steam loss of 5%. However, in a well-designed real plant, it can be expected that the steam loss will be close to zero. Reducing the steam loss has a direct effect on the product yield and on the thermal losses, especially of the dryer. The product yields, the total power demand and thermal losses at 5 and 0% steam loss are shown in Table 2.

In the reference case, the entire energy supply of the biorefinery is provided externally via electricity and natural gas. The torrefied biomass produced is materially utilised as output in the reference case. The resulting non-condensable volatile fraction of the torrefaction process is not thermally utilised in the reference case and is released to the atmosphere. Therefore, a second variant was investigated within the scope of this work, which operates self-sufficiently in terms of thermal energy. For this purpose, an incineration plant is added to the biorefinery, in which the entire torrefied biomass produced and the volatile fraction are thermally utilised. The Sankey diagram of this second variant is shown in Figure 5.

**Table 2.** Product yields, thermal and electrical power demand and losses for 5 and 0% steam loss (reference case).

| Output | 5% Steam Loss | 0% Steam Loss |
|---|---|---|
| Total thermal power demand | 499.61 kW | 499.79 kW |
| Total electrical power demand | 63.07 kW | 64.04 kW |
| Thermal loss drying | 7.54 kW | 0 kW |
| Thermal loss torrefaction | 80.64 kW | 77.70 kW |
| Torrefied biomass | 694.85 kg/h | 694.85 kg/h |
| 5-HMF | 51.9 g/h | 54.7 g/h |
| Furfural | 830 g/h | 873 g/h |
| Sodium formate | 1.254 kg/h | 1.32 kg/h |
| Sodium acetate | 13.46 kg/h | 14.17 kg/h |
| Methanol | 1.169 kg/h | 1.231 kg/h |

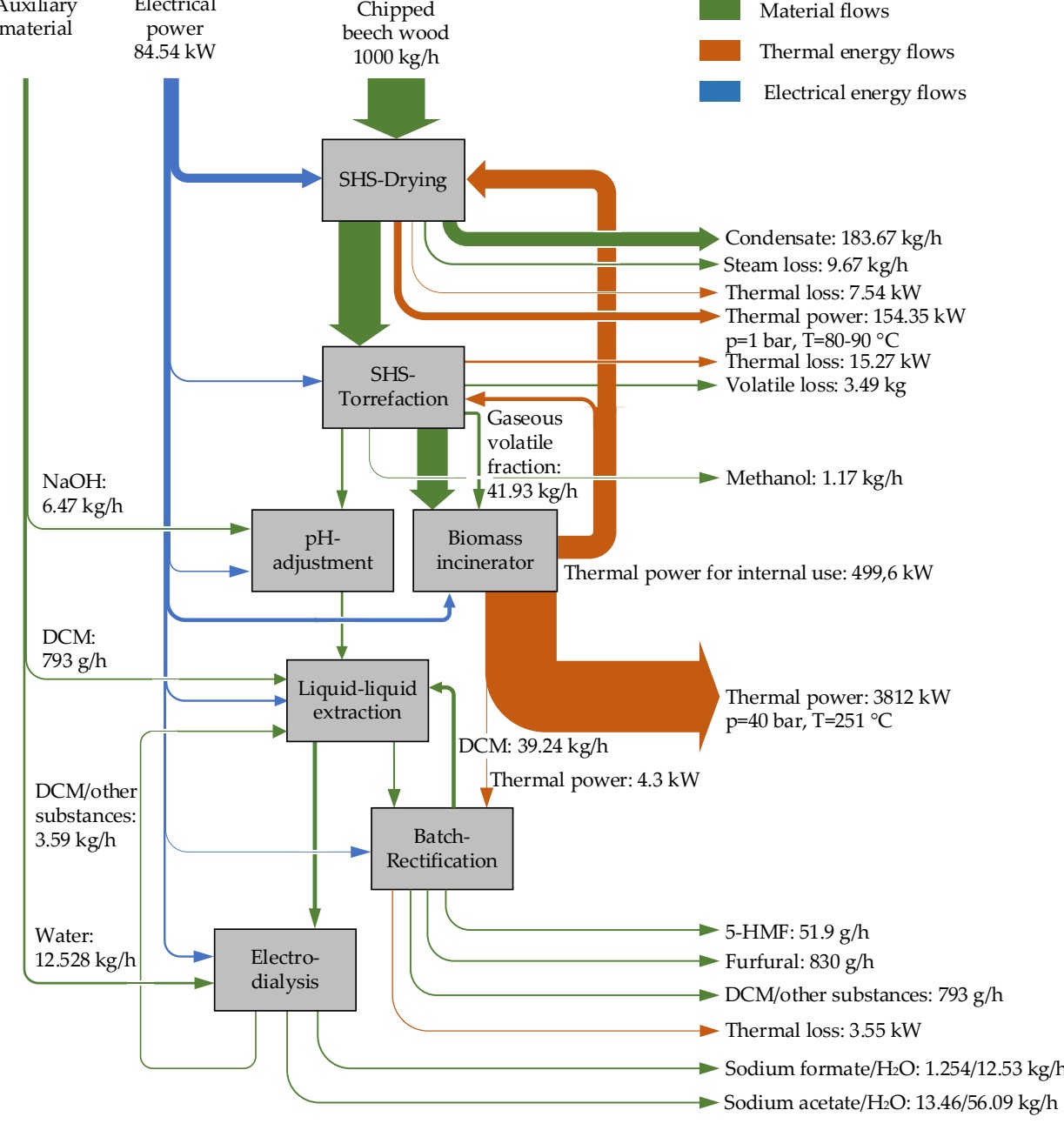

**Figure 5.** Energy and material input and output flows (incineration variant, 5% steam loss).

By combustion of the torrefied biomass and the volatile fraction, a thermal energy flow rate of approx. 4.3 MW can be generated assuming a calorific value of 22 MJ/kg. Since the plant's own thermal demand at 5% steam loss amounts to slightly less than 500 kW, approx. 3.8 MW of thermal power can be provided externally at 250 °C. In addition, 0.15 MW heat at 90 °C can be recovered from the SHS dryer. The reference case has a thermal loss of 92 kW, whereas the process variant has a thermal loss of only 26.36 kW. The process variant is particularly suitable for integration into existing, locally close production networks that have a heat demand at the temperature levels mentioned above. In terms of production volumes, the ratios at 5 and 0% steam loss in the process variant are the same as in the reference case. The process variant, however, causes higher electrical power consumption than the reference case, primarily due to the additional operation of a combustion fan and an exhaust gas fan for the combustion system. This additional consumption of electrical power totals approx. 21 kW.

### 4.2. Primary Energy Demand and $CO_2$ Emissions from a Process View

The energy embodied in natural resources prior to any human-made conversion is referred to as primary energy. The primary energy demand is the non-renewable portion of the primary energy required by the process to produce the end products; this is also referred to as cumulative non-renewable energy consumption.

The results presented here are based on a biomass input of 1000 kg per hour. The units 'kWh' and 'kgCO$_{2\text{-eq}}$' denote the total primary energy demand and $CO_2$ emission for processing 1000 kg of biomass per hour. The specific values of $CO_2$ emission and primary energy demand for the operation with 5% steam loss for the reference case and process variants are shown in this section. The results are categorized into three groups: upstream, thermal, and electrical. Upstream processes included the supply of auxiliary materials such as NaOH and DCM, as well as the supply chain of wood chips. Table 3 displays the values for the primary energy factor (PEF) and $CO_2$ footprint for the materials and energy carriers used in the process under investigation.

**Table 3.** The primary energy factor and $CO_2$ footprint of various materials and energy inputs.

| | Years | Primary Energy Factor (PEF) | | $CO_2$ Footprint | |
|---|---|---|---|---|---|
| NaOH [1] [27] | - | 6.11 [27] | kWh$_{PE}$/kg | 1.2 [1] | kgCO$_{2\text{-eq}}$/kg |
| DCM [1] | - | 11.26 | kWh$_{PE}$/kg | 3.42 | kgCO$_{2\text{-eq}}$/kg |
| Wood Supply chain [1] | - | 0.217 | kWh$_{PE}$/kg | 0.0547 | kgCO$_{2\text{-eq}}$/kg |
| Electricity [28,29] | 2018 | 1.71 | kWh$_{PE}$/kWh$_{el}$ | 0.471 | kgCO$_{2\text{-eq}}$/kWh |
| | 2019 | 1.55 | kWh$_{PE}$/kWh$_{el}$ | 0.408 | kgCO$_{2\text{-eq}}$/kWh |
| | 2020 | 1.37 | kWh$_{PE}$/kWh$_{el}$ | 0.366 | kgCO$_{2\text{-eq}}$/kWh |
| Natural Gas [28,30] | 2019 | 1.1 [30] | kWh$_{PE}$/kWh$_{el}$ | 0.202 [28] | kgCO$_{2\text{-eq}}$/kWh |

[1] Umberto calculation, ecoinvent 3.7.1.

Umberto LCA models were developed to calculate the total non-renewable energy consumption and $CO_2$ footprint of the upstream processes. Printouts of the models can be found as Supplementary Materials to this publication. Suitable activities were chosen from the Ecoinvent 3.7.1 database. These activities included all upstream activities, starting from the cradle and ending with the product's reception at the consuming entity, as well as the average transportation requirements. The region Europe (RER) was used for DCM, and the DE region with 'transport, freight, and lorry 16–32 metric ton EURO 4$'$ was chosen for wood chip supply. The global region had to be chosen for NaOH, and the calculated value was 19.33 MJ/kg. According to a report on sodium hydroxide eco-profiles, the gross primary energy required to produce 1 kg of NaOH is 22.04 MJ/kg (6.11 kWh/kg) [27]. This value is used in the calculations. The mass of the supplied product and the transportation requirements determine the upstream primary energy demand and $CO_2$ footprint. Because the average transportation distance of 100 km is already factored into the activities and the mass of biomass input is constant across all variants, the values of primary energy demand

(0.217 kWh/kg) and $CO_2$ emissions (0.0547 $kgCO_{2\text{-eq}}$/kg) are also constant. The required supply of auxiliary materials is determined by the percentage of steam/volatile loss during the process; the lower the volatile loss, the greater the mass supply, and thus the greater the primary energy demand and $CO_2$ footprint from these auxiliary materials.

Figures 6 and 7 depict the distribution of primary energy demand and $CO_2$ emissions to the individual processes, where 'others' refers to the sum of pH adjustment, liquid–liquid extraction, rectification, and electrodialysis processes. DCM and NaOH are separately indicated.

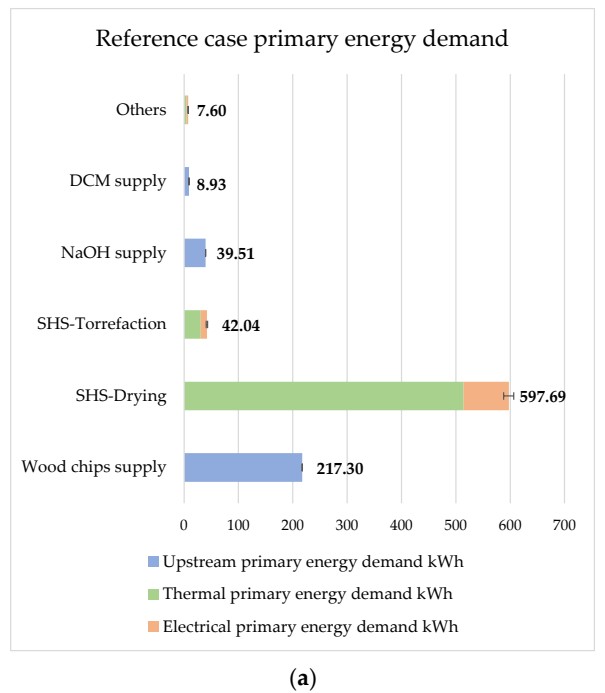
(**a**)

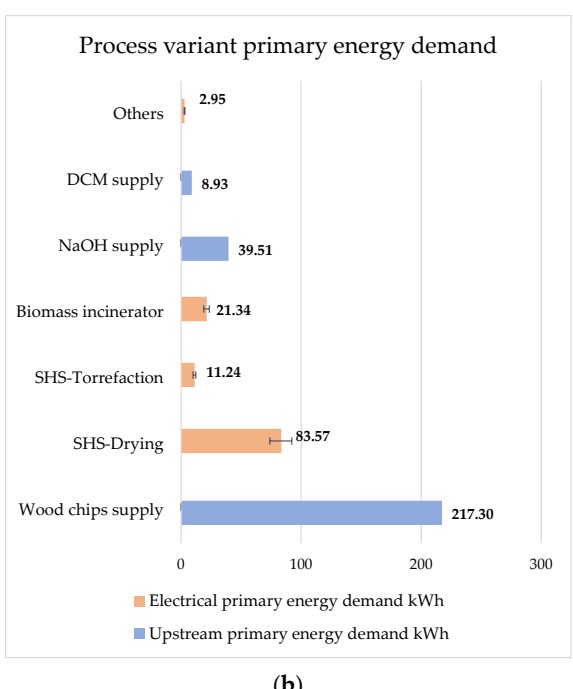
(**b**)

**Figure 6.** VALORKON primary energy demand with 5% steam loss: (**a**) reference case; (**b**) process variant.

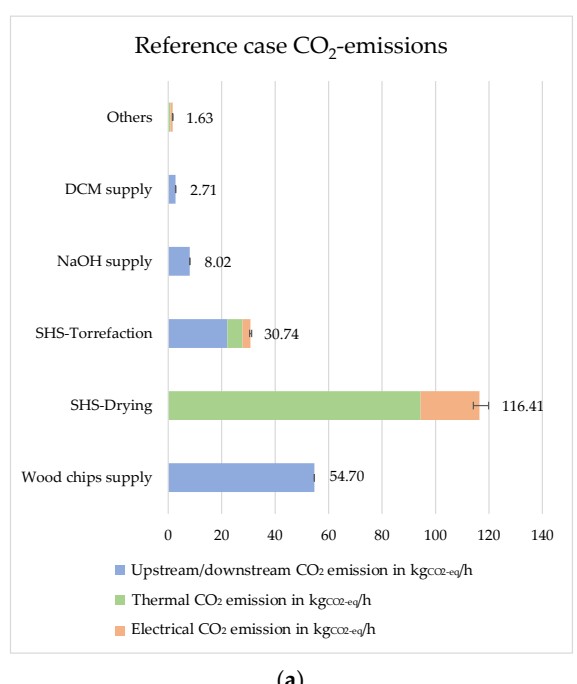
(**a**)

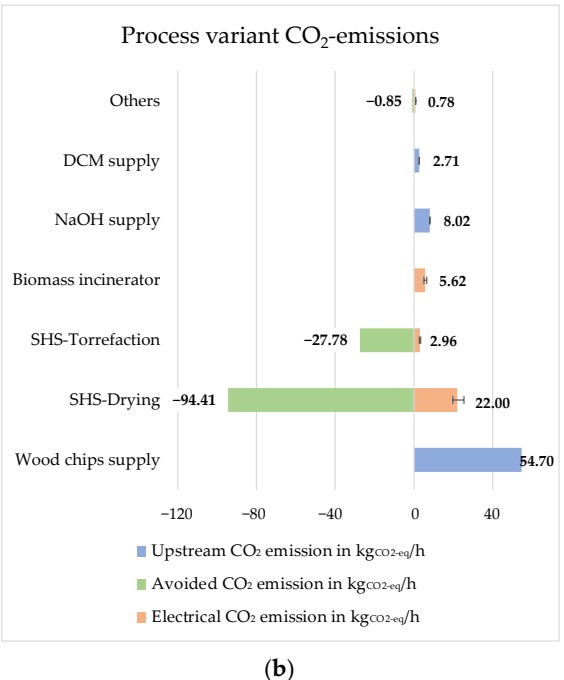
(**b**)

**Figure 7.** VALORKON $CO_2$ emission with 5% steam loss: (**a**) reference case; (**b**) process variant.

The thermal primary energy demand and $CO_2$ emissions were calculated by multiplying the PEF and $CO_2$ footprint of natural gas by the total amount of thermal energy required. Because the PEF of natural gas is constant (PEF = 1.1, where it includes the energy required for processing and distribution, i.e., 10%), as is the $CO_2$ footprint, there is no deviation in the bar charts for thermal primary energy demand and $CO_2$ emission. The electrical primary energy demand, on the other hand, is determined by the country's electricity mix. The German electricity mix for 2019 (prior to COVID-19) is considered here. The deviation shown in the black bar is derived from different PEF and $CO_2$ footprint values for the years 2018 and 2020, as shown in Table 3.

If the primary energy consumptions for processing 1000 kg of biomass per hour for both variants are added up counting 5% steam loss, the result is 913.07 kWh for the reference case and 384.84 kWh for the process variant. The significant difference in primary energy demand is due to the fact that in the process variant the required thermal energy is provided internally by a biomass incinerator. Figure 6 also shows the distribution of primary energy demand for both variants when varying the primary energy demand for the individual process steps according to Table 3. The SHS-drying process consumes the most primary energy in the reference case, followed by the biomass supply chain, whereas the biomass supply chain consumes the most primary energy in the process variant, followed by the SHS drying. The process variant includes an additional electrical energy consumer, a biomass incinerator with a primary energy demand of 21.34 kWh per hour.

Figure 7 illustrates the $CO_2$ emissions and their distribution due to the variation of the primary energy factors for electric power according to Table 3 for the different process stages and the supply chain. During the torrefaction process, the non-condensable gaseous volatile fraction, which also contains methane, is released into the environment in the reference case and fed to a biomass incinerator in the process variant case. Since methane is released into the environment, the corresponding downstream $CO_2$ equivalents in the torrefaction process must be added for the reference case. For this purpose, a $CO_2$ equivalence factor of 25 was applied according to [31]. Total $CO_2$ emissions are 214.21 kgCO$_{2\text{-eq}}$ per ton of biomass input for the reference case and 96.78 kgCO$_{2\text{-eq}}$ per ton of biomass input for the process variant. The calculation with Umberto resulted in $CO_2$ emissions of 218.55 kgCO$_{2\text{-eq}}$ per ton of biomass input for the reference case, thus verifying the calculations. Since the biomass incinerator provides the necessary thermal energy for the process, an emission of 123.04 kgCO$_{2\text{-eq}}$ per hour is avoided, which also includes the avoided $CO_2$ emission from the combustion of non-condensable gaseous volatile fraction. The negative side of the bar chart represents avoided $CO_2$ emissions, while the positive side represents caused $CO_2$ emissions in the process variant. The avoided $CO_2$ emissions are calculated under the assumption that the combustion of natural gas is replaced in the biorefinery. For the reference case, the SHS-drying process emits the most $CO_2$, followed by the wood supply chain, and for the process variant, drying has the second highest caused $CO_2$ emission but avoids most $CO_2$ emission through the internal heat supply.

*4.3. Primary Energy Demand and $CO_2$ Emissions from a Product View (Allocation)*

To know how much primary energy is required or how much $CO_2$ is emitted to produce one kilogram of the respective end product, the total primary energy demand and $CO_2$ emission must be allocated to the end product. To begin, the end products are set mass based in relation to the total output of a respective sub-process. The primary energy demand/$CO_2$ emissions of the individual processes can then be allocated to the corresponding end products. The total primary energy requirement/$CO_2$ emissions to produce the respective end product can then be calculated by adding the individual values. This can be divided by the respective product output to obtain the primary energy demand/$CO_2$ emission per kg of end product.

Figure 8a represents a comparative representation of the primary energy demand required to produce one kilogram of the respective end product for both variants. It clearly shows for the reference case that the products 5-HMF and furfural have the highest primary

energy requirement per kilogram, with 9.6 kWh each, followed by sodium formate and sodium acetate, with 4.5 kWh each. It can be seen that the primary energy demand for the process variant is significantly lower because the required thermal energy is covered by the incinerator; the same is true for $CO_2$ emission. Figure 8b represents for both variants the $CO_2$ emissions that occur when producing one kilogram of end product. The highest emission per kilogram occurs when producing 5-HMF and furfural for the reference case, and when producing sodium acetate and sodium formate for the process variant. The effect of the process variant is more significant on 5-HMF and furfural, because the thermal energy requirement during rectification has a significant weight on 5-HMF and furfural, and because this can be supplied internally in the process variant, the value of primary energy demand and $CO_2$-emission is further reduced. The higher $CO_2$ emission and primary energy demand of furfural, 5-HMF, sodium acetate, and sodium formate can be explained by the small amount of product produced and the number of steps required to obtain these products. Torrefied beech wood, on the other hand, emits the least amount of $CO_2$ per kilogram. This is due to the small amount of beech wood input required to produce one kilogram of torrefied beech wood.

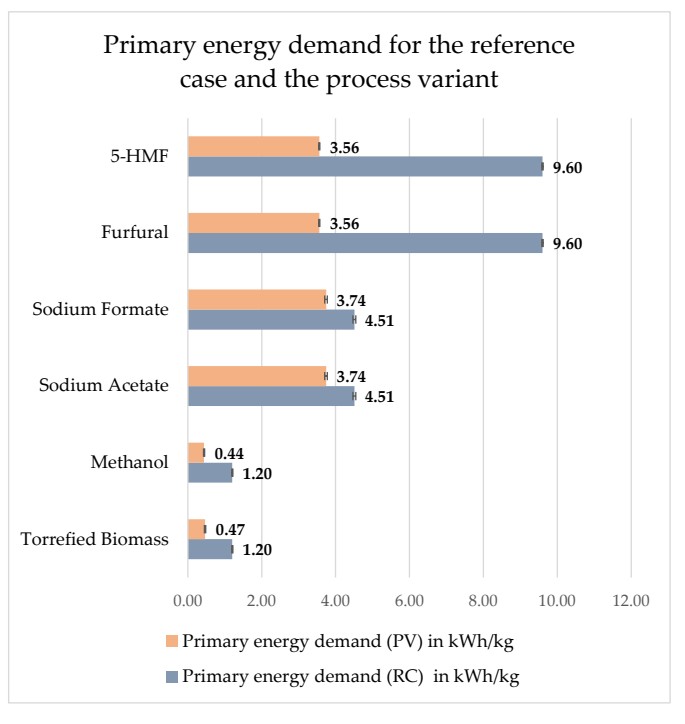

(**a**)

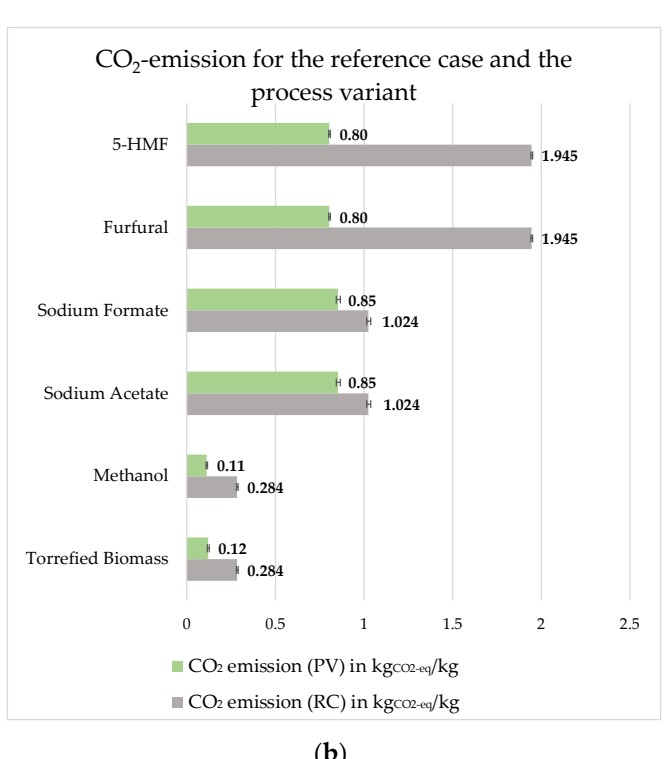

(**b**)

**Figure 8.** Comparison from product perspective with 5% steam loss: (**a**) primary energy demand for the reference case and the process variant; (**b**) $CO_2$ emission for the reference case and the process variant.

When the steam loss is reduced to 0%, it has a minor effect on the reference case and almost no effect on the process variant's primary energy demand and $CO_2$ emissions. The primary energy demand and $CO_2$ emissions for the reference case with 0% steam loss are shown in the Table 4 below, where the value of primary energy demand for furfural and 5-HMF has been reduced to 9.55 kWh/kg, while the values for other products have remained the same as with 5% steam loss. The $CO_2$ emissions, on the other hand, have been slightly reduced for all products.



**Table 4.** Primary energy demand and $CO_2$ emission for the reference case with 0% steam loss.

| Products | Primary Energy Demand | $CO_2$ Emission |
|:---:|:---:|:---:|
| | kWh/kg | $kgCO_{2\text{-eq}}$/kg |
| Torrefied biomass | 1.20 | 0.25 |
| Methanol | 1.20 | 0.25 |
| Sodium acetate | 4.51 | 0.99 |
| Sodium formate | 4.51 | 0.99 |
| Furfural | 9.55 | 1.90 |
| 5-HMF | 9.55 | 1.90 |

*4.4. Further Environmental Impacts due to Changes in the Biomass Feedstock*

As outlined in the methodology chapter, it was investigated for the reference case how the conversion of the biomass input from beech wood chips to straw residues affects the environmental impacts. For this purpose, the Umberto model on the reference case was used to investigate which other environmental indicators change significantly when switching to a biomass residue originating from intensive agriculture. Only the supply chain for the biomass was altered, but not the product streams obtained, as their concentrations had so far only been investigated experimentally using beech wood. This analysis showed that the following categories in particular are changing strongly:

- Climate change, GWP 100 a in $kgCO_{2\text{-eq}}$
- Human toxicity w/o LT, HTPinf w/o LT in kg 1,4-$DCB_{\text{-eq}}$
- Freshwater eutrophication w/o LT, FEP w/o LT in kg $P_{\text{-eq}}$
- Freshwater ecotoxicity w/o LT, FETPinf w/o LT in kg 1,4-$DCB_{\text{-eq}}$

The results represented here are based on 1000 kg of biomass input of the two biomass residues considered. Figure 9a shows the global warming potential. The methodology used for the impact assessment was IPCC 2013, which takes the impacts of emissions over a period of 100 years into account. The total emission for the beechwood biomass is 218.55 $kgCO_{2\text{-eq}}$. When using straw residues, the total emission is 254.73 $kgCO_{2\text{-eq}}$. The additional 36.18 $kgCO_{2\text{-eq}}$ is caused by the straw supply chain, which has a $CO_2$ footprint of 0.0908 $kgCO_{2\text{-eq}}$/kg of straw, whereas the beechwood has a $CO_2$ footprint of 0.0547 $kgCO_{2\text{-eq}}$/kg of beechwood.

Figure 9b depicts human toxicity, freshwater ecotoxicity and freshwater eutrophication potential for the two investigated biomass types. For the impact estimation method, ReCiPe Midpoint (H) w/o LT [26] has been used. All substances which are considered to be toxic are standardized at 1,4-dichlorobenzene (DCB). Total emissions from substances toxic to humans from the reference case are 7.729 kg 1,4-$DCB_{\text{-eq}}$ for beechwood and 23.599 kg 1,4-$DCB_{\text{-eq}}$ for straw, while total emissions having an ecotoxic impact on freshwater ecosystems are 0.079 kg 1,4-$DCB_{\text{-eq}}$ for beechwood and 3.103 kg 1,4-$DCB_{\text{-eq}}$ for straw. All eutrophication-potential substances are converted to the same amount of phosphorous (P) with the same eutrophication impact. The total accumulation of excess nutrients in a body of water is 0.010 kg $P_{\text{-eq}}$ for beechwood and 0.014 kg $P_{\text{-eq}}$ for straw residues.

Agricultural cultivation is associated with considerable environmental burdens due to the use of pesticides, fertilisers, machinery and increased water consumption, among other things. Since agricultural cultivation is responsible for most of the impact, beech wood as a biomass residue input has a lower environmental impact than straw residues.

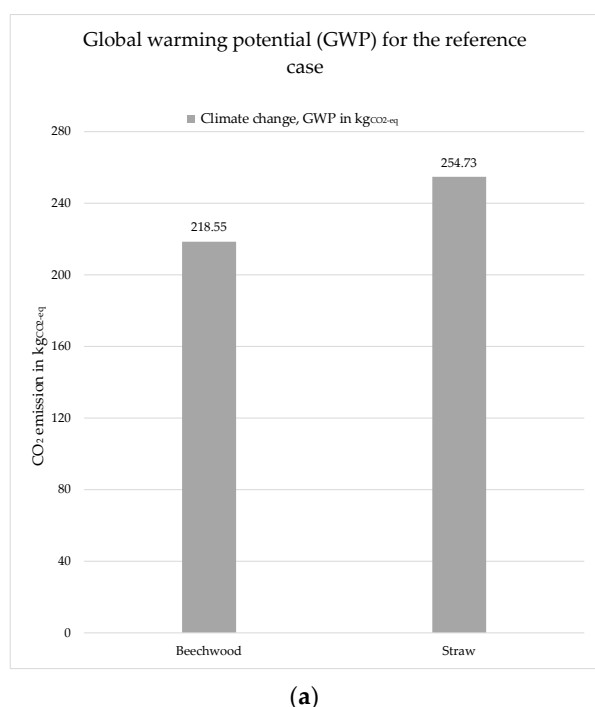

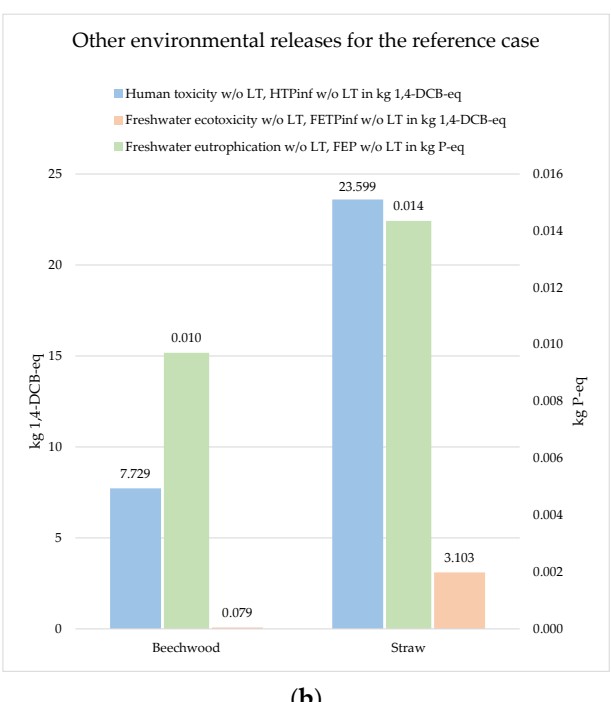

(**a**)            (**b**)

**Figure 9.** Life cycle impact assessment of the reference case for different biomass inputs: (**a**) global warming potential (GWP); (**b**) other environmental releases.

### 4.5. Sensitivity Analysis

According to reference [32], the HHV of torrefied biomass ranges from 22 to 28 MJ/kg due to an increase in carbon content. The heat generated during biomass incineration and the thermal power available to external customers vary depending on the HHV of torrefied biomass. Figure 10a shows a sensitivity analysis concerning the HHV of torrefied biomass. The blue line represents the excess thermal power that could be sold to an external customer, and the green line represents the avoided $CO_2$ emissions from supplying excess thermal energy in the process variant, assuming that it replaces natural gas consumption. Within the scope of this study, an HHV of 22 MJ/kg was considered. This value can be used as a reference value in the diagram, and all process variant calculations are based on it. At this value, the excess thermal power and avoided $CO_2$ emissions are 3.81 MW and 769 $kgCO_2$. From this reference point, an elevation of 2 MJ/kg of HHV increases thermal power and $CO_2$ emissions by 10.12%.

Another important factor that must be considered is the moisture content of the input biomass. The moisture content of biomass varies depending on the type of biomass and amount of time it has been stored. The SHS dryer contributes to the majority of the overall biorefinery's primary energy demand. The latter is highly sensitive to the moisture content of biomass, as shown in Figure 10b, where the moisture content, based on dry matter ranges from 20 to 45%. These changes would have an impact on the total energy demand as well as the $CO_2$ emissions. As shown in the diagram, the effect of this parameter on both variants is different. In the reference case, the SHS dryer's total primary energy demand includes both electrical and thermal primary energy demand. The SHS heater is the main energy consumer, and as the moisture content decreases, it requires less superheated steam supply to dry the biomass input, resulting in a decrease in energy consumed by the SHS heater. In the case of a process variant, however, the total primary energy demand consists solely of the electrical primary energy demand. The steam fan is the largest electrical energy consumer, accounting for 96% of the total electrical primary energy demand. As the amount of steam decreases with decreasing moisture content, so does the energy required by the fan to recirculate the steam. The basic calculation was carried out for a moisture content of 35%.

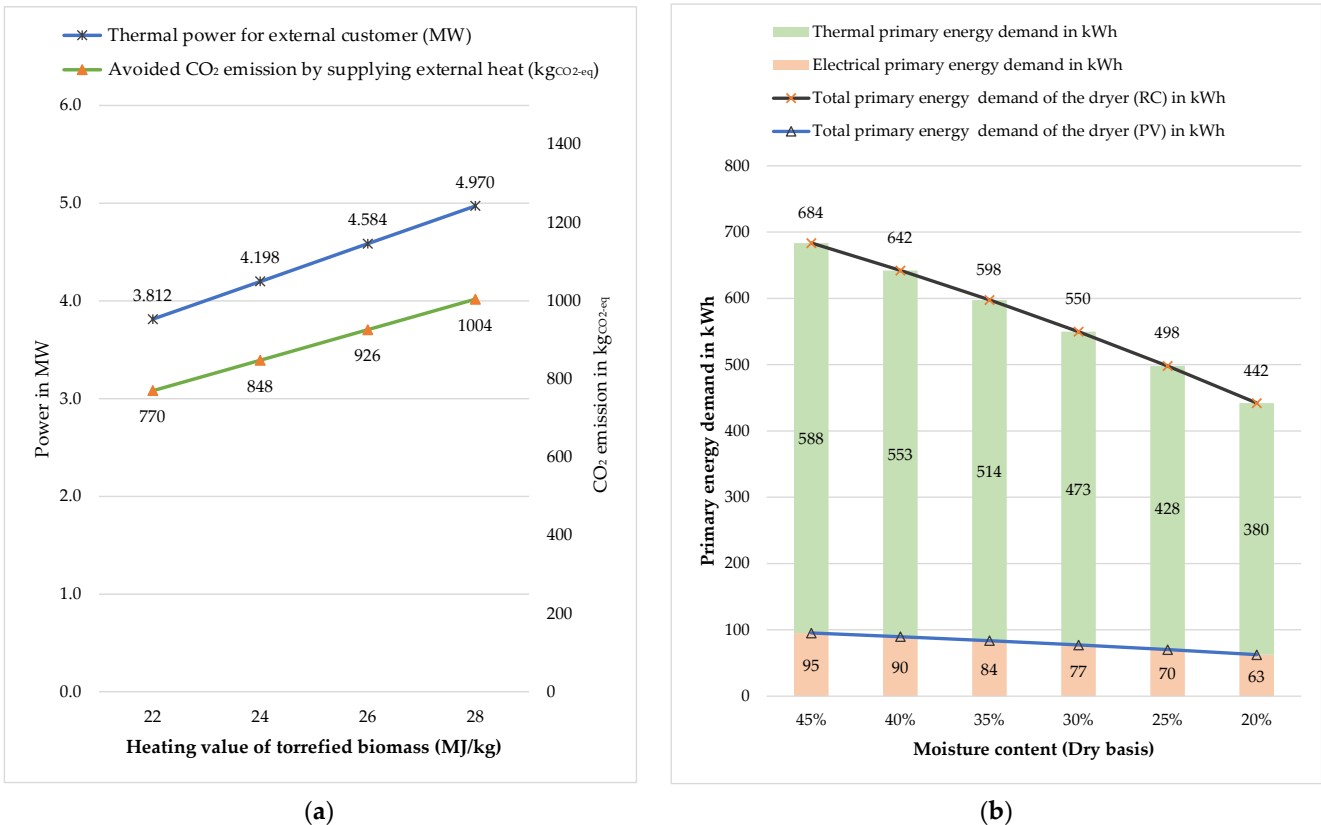

**Figure 10.** Sensitivity analysis: (**a**) heat value of torrefied biomass; (**b**) moisture content of biomass (dry basis).

The composition of electricity generation, the so-called electricity mix, was related to Germany in the basic calculations (see also Table 3). The electricity mix is not a constant factor. The electricity mix depends on the deployment of renewable energy sources and thus is country and time dependent. Figure 11a shows the $CO_2$ emissions for both variants with different electricity mixes. This graph can be used to estimate total $CO_2$ emissions for various countries, as well as in the future when the electricity mix changes.

The biomass supply chain is divided into two parts: wood harvesting/sawmill and transport. An LCA model was created to calculate the cumulative non-renewable energy demand or primary energy demand and $CO_2$ emissions of the biomass supply chain, which included the transportation activity "transport, freight, lorry 16–32 metric ton EURO 4 [RER]". The primary energy demand and $CO_2$ emission due to harvesting and sawmilling are constant, as the study is based on 1000 kg of biomass processing, and the values are 143.61 kWh and 38.17 $kgCO_{2\text{-eq}}$, respectively. Hence, the total primary energy demand and $CO_2$ emission of the biomass supply chain are primarily determined by the supply distance. Figure 11b shows the variation in primary energy demand and $CO_2$ emissions based on the supply distances ranging from 100 to 300 km. The reference distance is 100 km, resulting in a primary energy demand of 217.13 kWh and $CO_2$ emission of 54.7 $kgCO_{2\text{-eq}}$ for the reference case. From the reference point, an increase of 50 km increases the primary energy demand by 17.16% and $CO_2$ emission by 15.07%.

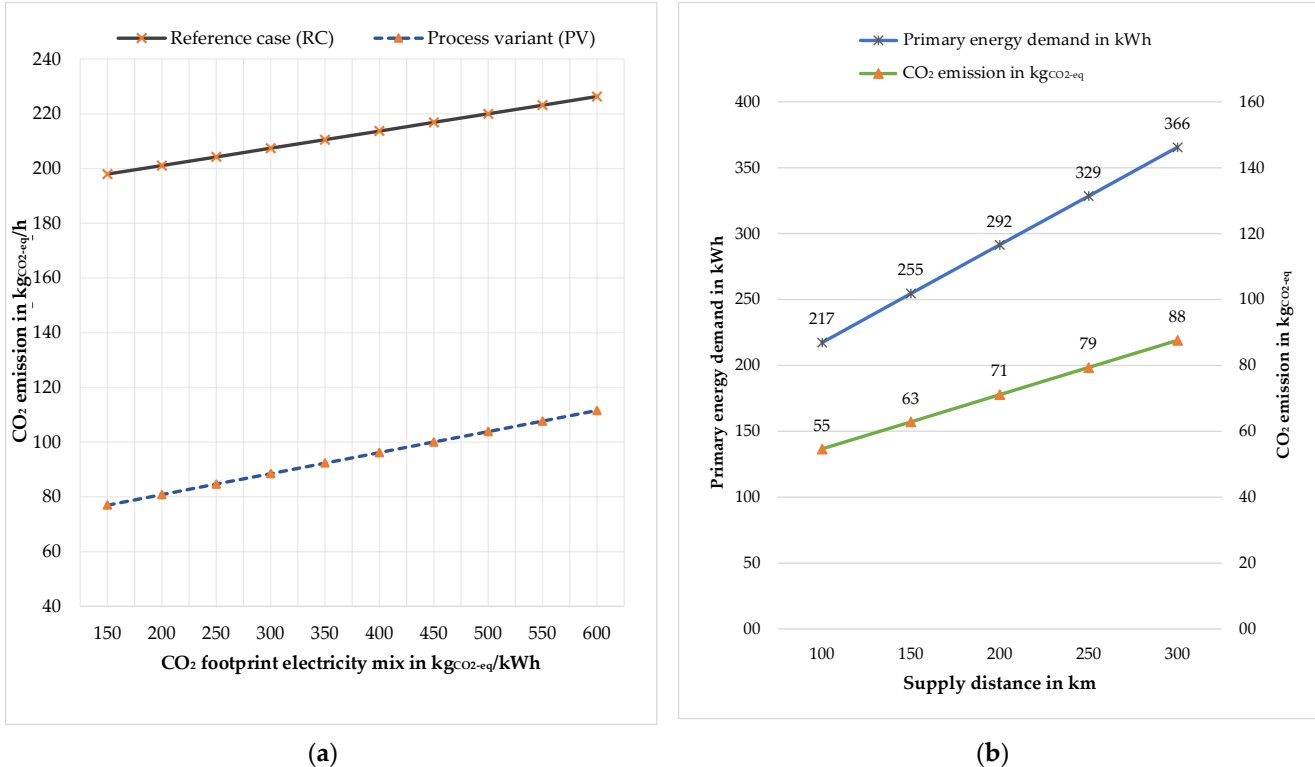

(**a**)  (**b**)

**Figure 11.** Sensitivity analysis: (**a**) $CO_2$-emission for the reference case and process variant with different electricity mix; (**b**) biomass supply distance.

## 5. Discussion

As part of the research project to develop the biorefinery presented here, competing processes for the production of 5-HMF and furfural were also investigated. In reference [6], the results of a first investigation of the process of reference [33] for the production of 5-HMF were presented. The process under investigation was at the laboratory stage. The total $CO_2$ emissions determined ranged from 326 to 1160 $kgCO_2/kg_{HMF}$, with the main contributor being the use of dichloromethane [6]. In the meantime, AvaBiochem, in close cooperation with the Karlsruhe Institute of Technology, has brought another process for the production of 5-HMF to market maturity [34,35]. For this process, $CO_2$ emissions between approx. 9 and 49 $kgCO_2/kg_{HMF}$ (depending on the degree of energy recovery and the energy source used) were determined in this project. The newly developed biorefinery presented in this publication is ecologically advantageous compared to the competing processes with 1.945 $kgCO_2/kg_{HMF}$ for the reference case and 0.80 $kgCO_2/kg_{HMF}$ for the variant with biomass combustion. The supply chains are considered in all studies.

For furfural, investigations by reference [6] identified a range of approx. 13 to 14 $kgCO_2/kg_{furfural}$ for the Huaxia-Westpro process. Here, all input chains were included except for the biomass raw material. The main $CO_2$ driver here is the provision of heat for the production process, accounting for approx. 90%. Further investigations in this research project concerning that process showed that the $CO_2$ emissions of the Huaxia-Westpro process can be reduced to 1.6 to 3.7 $kgCO_2/kg_{furfural}$ by heat recovery and switching to energy sources with lower $CO_2$ emissions. The upper value refers to production in China. These values include all input supply chains and transports, including those of the biomass raw material maize. The allocated $CO_2$ emission values for the new biorefinery presented here are at the same level as the values mentioned above for 5-HMF, because the same process steps have to be passed through for the production of both platform chemicals. In the reference case, the $CO_2$ emissions of the investigated biorefinery (1.945 $kgCO_2/kg_{furfural}$) are slightly above the best value of the Huaxia-Westpro process. In the process variant

with its own thermal energy supply, the new biorefinery causes less than 50% of the $CO_2$ emissions of the competing process. Here, too, all preceding supply chains are considered.

Sodium acetate and sodium formate are follow-on products from the reaction of acetic acid with sodium hydroxide. For conventional fossil-based acetic acid production, the authors of reference [36] give a value of 1.846 $kgCO_2/kg_{acetic\ acid}$. The authors' own examination of the upstream chains of methanol and carbon monoxide with natural gas as raw material results in 0.89 $kgCO_2/kg_{acetic\ acid}$ for methanol ([36]: 0.36 $kgCO_2/kg_{acetic\ acid}$) and 0.994 $kgCO_2/kg_{acetic\ acid}$ for carbon monoxide ([36]: 1.065 $kgCO_2/kg_{acetic\ acid}$). For classical acetic acid production, e.g., by the Celanese company in Texas, $CO_2$ emissions of approx. 2.3 $kgCO_2/kg_{acetic\ acid}$ can therefore be assumed. For sodium acetate and sodium formate combined, the biorefinery investigated in this study produces a total of 2.048 $kgCO_2/kg$ for the reference case and 1.7 $kgCO_2/kg$ for the process variant with thermal auto-supply. The deduction of the $CO_2$ proportion of the sodium hydroxide solution used for comparison with acetic acid is omitted here because the new biorefinery does not cause higher specific $CO_2$ emissions than the classic competing process based on natural gas.

This comparison applies to operation with beech wood chips with a humidity content of 35% (dry). If the moisture content can be further reduced through natural solar drying and storage without using non-renewable energy carriers, the $CO_2$ emissions of the new biorefinery will decrease further. The above comparisons do not include $CO_2$ credits from the sale of excess thermal energy and thus the external substitution of fossil energy sources, such as natural gas.

The $CO_2$ emissions determined are mainly based on the use of fossil non-renewable raw materials (except limestone) and thus also give a reflection of the energy demand of the processes considered. The newly developed biorefinery is thus competitive from an ecological and energetic point of view compared to the bio-based competing processes for the production of 5-HMF and furfural and the classic natural gas-based process for the production of acetic acid.

The laboratory tests with the batch-rectification column have shown a loss of dichloromethane of approx. 1% or less. However, a loss of 2% was conservatively assumed for the calculations. Accounting for 2.71 $kgCO_{2\text{-}eq}/h$, dichloromethane causes 1.27% of the total $CO_2$ emissions in the reference case and 2.8% in the process variant with thermal self-supply. If the dichloromethane losses in a commercial plant were similar to those in the laboratory tests, the $CO_2$ emissions caused by the consumption of dichloromethane would be cut in half.

The biorefinery's steam loss has almost no impact on total energy consumption and total $CO_2$ emissions. However, the steam loss affects the product yield and thus also the consumption of sodium hydroxide solution and dichloromethane. The steam loss was assumed to be 5%, but given the design conditions and the associated phase separation, it should be between 0 and 1% for a well-designed system. The increased product yield and the additional consumption of auxiliary materials can be extrapolated straightforwardly from the values shown in Figures 4 and 5. The product-related, allocated $CO_2$ emission values are reduced accordingly. Reducing the steam losses only marginally changes the overall picture presented so far.

Thus, the moisture content of the biomass represents the greatest lever for reducing energy demand and $CO_2$ emissions if the reduction of the humidity level is achieved in a natural way, e.g., by dry storage for several months or as described above. A reduced moisture content reduces the demand for thermal energy but also for electrical energy to drive the fan to circulate the superheated steam.

However, the transport distance for the biomass raw material also has a significant influence on the primary energy demand and $CO_2$ emissions. The baseline calculations were carried out for a distance of 100 km. Distances up to 300 km were considered in the sensitivity analysis. Since the plant concept under consideration is based on relatively small

input quantities, the objective should be, from an environmental point of view, to obtain the biomass residue as locally as possible, with a distance of well under 100 km.

The nature and the quantity of the expected compounds in the condensable volatile fraction depend mainly on the composition (cellulose, hemicellulose and lignin) of the biomass [11]. Examples of promising feedstock are: hay, straw, wood chips, digestate, manure or other bio-based residues. Orive et al. (2020) showed that olive residues are a promising feedstock for the extraction of high-value chemicals [37].

The process parameters (temperature, residence time and flow velocity) and the particle size of the used feedstock have a great influence on the process liquid and solid fraction outputs [19,38]. The process parameters (temperature and residence time) have been selected based on experimental data in order to guarantee a good compromise between obtained liquid and solid fractions.

A mass-based allocation of the primary energy demand and the $CO_2$ emissions to the generated products was shown as an example. A value-based allocation would be possible in principle. However, since the monetary values of the products can change dynamically depending on demand, availability and product quality, and the biorefinery can extract further valuable materials, a value-based allocation was not carried out in the context of this work.

Apart from the possibility of supplying thermal energy via the combustion of torrefied biomass, no other process integration scenarios were considered within the scope of this work. The biorefinery could, for example, be integrated into an industrial symbiosis in which electrical power is generated in an environmentally friendly manner in addition to thermal energy. The pulp and paper industry, the chemical industry, the pharmaceutical industry and the steel producing industry, for example, would be suitable for such a symbiosis. It would also be conceivable to further process the torrefied biomass into activated carbon or to generate syngas.

Commercially, torrefaction is at an early stage of development. Several technology companies are aiming for a commercial launch, but are struggling with technical problems in demonstration plants. Non-oxidative torrefaction has higher commercialization potential compared to other developed methods because the yield is higher and the process operation is safer. The information currently available on the practical applications of biomass torrefaction in industry is still insufficient [11].

The transition to a bioeconomy is increasingly being demanded and supported politically. In 2018, the EU Commission presented an action plan for the development of a sustainable and cycle-oriented bioeconomy which, among other factors, provides considerable financial resources for the establishment of biorefineries in Europe [39]. In their strategy development, companies should analyse potential conflicts of objectives with regard to ecological, economic and social sustainability at an early stage. Ecological conflicts of interest should be discussed here first, because ecological sustainability is the primary claim with which bio-based products and the bio-economy as an economic system are promoted [4]. The paper presented here clearly demonstrates that the newly developed biorefinery can make a positive contribution on the way to climate neutrality from an ecological point of view.

## 6. Conclusions and Outlook

In the context of this work, the energy demand, $CO_2$ footprint and other environmental parameters were analysed for a newly developed biorefinery, which works on the principle of torrefaction with superheated steam. The whole supply chain of biomass and auxiliary materials were included in the assessment. The results show that the biorefinery can compete with established bio-based and conventional production processes for furfural, 5-HMF and acetic acid in terms of its environmental performance indicators. The biorefinery is versatile enough to produce various high-quality platform chemicals and torrefied biomass from biomass residues.

The main drivers for further optimisation were identified. In this context, the plant is particularly interesting for an industrial symbiosis with actors from several different sectors. The biorefinery particularly supports the EU's 2050 goal of climate neutrality by using biomass residues in an environmentally friendly way to produce high-quality platform chemicals.

The research has shown that it is very time consuming to collect reliable data for the environmental analysis, especially for the competing processes. Wherever possible, the results were verified by means of the authors' own experiments, calculations and estimates.

In the second part of this publication series, the economic performance and possible commercialisation strategies are presented. Further works will focus on the continuous separation of chemicals, the flexible extraction of different valorisation products and the upscaling of the SHS-based drying/torrefaction with different biomass residues.

**Supplementary Materials:** The following are available online at https://www.mdpi.com/article/10.3390/su14031212/s1: Umberto-models: Roy-etal-MDPI-2022-Umberto-model-Reference-Case-Beechwood.jpg, Roy-etal-MDPI-2022-Umberto-model-Process-Variant-Beechwood.jpg.

**Author Contributions:** Conceptualization, B.R., P.K.-M. and A.D.; methodology, P.K.-M. and A.D.; software, B.R.; validation, B.R.; formal analysis, B.R., P.K.-M. and A.D.; investigation, B.R.; resources, P.K.-M.; data curation, B.R.; writing—original draft preparation, B.R. and P.K.-M.; writing—review and editing, P.K.-M. and A.D.; visualization, B.R. and P.K.-M.; supervision, P.K.-M. and A.D.; project administration, P.K.-M. and A.D.; funding acquisition, P.K.-M. and A.D. All authors have read and agreed to the published version of the manuscript.

**Funding:** This research was funded by the FEDERAL MINISTRY OF EDUCATION AND RE-SEARCH (BUNDESMINISTERIUM FÜR BILDUNG UND FORSCHUNG), grant number 031B0664.

**Institutional Review Board Statement:** Not applicable.

**Informed Consent Statement:** Not applicable.

**Data Availability Statement:** Data are contained within the article or Supplementary Materials.

**Conflicts of Interest:** The authors declare no conflict of interest.

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
