# Peer review of "Superheated Steam Torrefaction of Biomass Residues with Valorisation of Platform Chemicals—Part 1: Ecological Assessment"

_sustainability, doi:10.3390/su14031212_

Round 1

Reviewer 1 Report

The manuscript is about life cycle analysis of feedstock using a novel pilot plant in Germany. It is well written and structured and has only minor recommendations to improve the manuscript.

(1) Could the authors also include the governance and policies in the discussion of this manuscript?

(2) Discuss also why you were focused on straw and beechwood in the analysis, willow farms and waste feedstocks could be and should be used. Olive stones and wood chips are of strong value.

(3) Do authors see potential to prepare domestic stove briquettes from their torrefaction feedstocks? Characterization of woodstove briquettes from torrefied biomass and coal 

(4) I would not write anything about pharma and paper industry. These industries have less interested in torrefaction and more in hydrolysis because there is an option to win projects of even higher value than you have presented in your manuscript. 

(5) Can you also include in the discussion the selected operating parameters and especially the particle size and which technical problems stop industry from the use of torrefaction? The only one commercialized pilot plant in Europe that produces commercialized products is Arigna Fuels plant in Ireland.

Reviewer 2 Report

This paper is interestiong for other researchers. It is well written and well discussed. I Suggest only some minor corrections.

*Lines 118-124: please put the goal of the paper only in introduction section.

*Assumptions of Table 1: Please add some references you used to propose these assumptions. You can add some authors that also use similar assumptions.

Reviewer 3 Report

This article is part of a greater research in the field of energy, with particular focus on biomass and in detail on a specific innovative process which refers to superheated steam drying and torrefaction of biomass residues. 

The article is well prepared, with a technical approach which however is presented in such manner that a large category of readers can understand the content.

Clearly explained methodology.

Really good discussion, comprising elements of some other current works on the same subject, presenting and discussing results in accordance with other relevant researches. 

I have a single recommendation: the authors should try to mention the difficulties of developing such a research, or the limits of the study in the final section of the article.

Good luck with your future research!
